# Mosaicking to Distill:
# Knowledge Distillation from Out-of-Domain Data

**Gongfan Fang**[1,4]**, Yifan Bao**[1]**, Jie Song**[1]**, Xinchao Wang**[2]**, Donglin Xie**[1]
**Chengchao Shen**[3]**, Mingli Song**[1][*]
[1]Zhejiang University, [2]National University of Singapore, [3]Central South University
[4]Alibaba-Zhejiang University Joint Institute of Frontier Technologies
{fgf,yifanbao,sjie,donglinxie,brooksong}@zju.edu.cn
xinchao@nus.edu.sg, scc.cs@csu.edu.cn

## Abstract

Knowledge distillation (KD) aims to craft a compact student model that imitates the behavior of a pre-trained teacher in a target domain. Prior KD approaches, despite their gratifying results, have largely relied on the premise that *in-domain* data is available to carry out the knowledge transfer. Such an assumption, unfortunately, in many cases violates the practical setting, since the original training data or even the data domain is often unreachable due to privacy or copyright reasons. In this paper, we attempt to tackle an ambitious task, termed as *out-of-domain* knowledge distillation (OOD-KD), which allows us to conduct KD using only OOD data that can be readily obtained at a very low cost. Admittedly, OOD-KD is by nature a highly challenging task due to the agnostic domain gap. To this end, we introduce a handy yet surprisingly efficacious approach, dubbed as *MosaicKD*. The key insight behind MosaicKD lies in that, samples from various domains share common local patterns, even though their global semantic may vary significantly; these shared local patterns, in turn, can be re-assembled analogous to mosaic tiling, to approximate the in-domain data and to further alleviating the domain discrepancy. In MosaicKD, this is achieved through a four-player min-max game, in which a generator, a discriminator, a student network, are collectively trained in an adversarial manner, partially under the guidance of a pre-trained teacher. We validate MosaicKD over classification and semantic segmentation tasks across various benchmarks, and demonstrate that it yields results much superior to the state-of-the-art counterparts on OOD data. Our code is available at https://github.com/zju-vipa/MosaicKD.

## 1 Introduction

Knowledge distillation (KD) has emerged as a popular paradigm for model compression and knowledge transfer, attracting attention from various research communities [18, 41, 47, 15]. The goal of KD is to train a lightweight model, known as the student, by imitating a pre-trained but more cumbersome model, known as the teacher, so that the student masters the expertise of the teacher. In recent years, KD has demonstrated encouraging results over various machine learning applications, including but not limited to computer vision [6, 29], data mining [2], and natural language processing [46, 20]

Nevertheless, the conventional setup for KD has largely relied on the premise that, data from at least the same domain, if not the original training data, is available to train the student. This seemingly-mind assumption, paradoxically, imposes a major constraint for conventional KD approaches: in

---

[*]Corresponding author.

35th Conference on Neural Information Processing Systems (NeurIPS 2021).

many cases, the training data and even their domain for a pre-trained network are agnostic, due to for example confidential or copyright reasons. Hence, the in-domain prerequisite significantly limits the applicable scenarios of KD, and precludes taking advantage of the sheer number of publicly-available pre-trained models, many of which with unknown training domain [25, 39], to carry out massive knowledge transfer.

In this paper, we aim at the ambitious goal of conducting KD using only *out-of-domain* (OOD) data, which, in turn, enables us to greatly relax the conventional prerequisite and thereby largely strengthens applicability of KD. Unarguably, OOD-KD is by nature a highly challenging task, since the domain discrepancy will inevitably impose a major obstacle towards the proper functioning of the pre-trained teacher. In fact, if we are to conduct naive KD on the raw OOD data, the resulting student model, as will be demonstrated in our experiments, fails to provide any performance guarantee on the target domain. This phenomenon signifies the limited generalization capability learned from OOD data, which is unsurprising.

To this end, we propose a novel assembling-by-dismantling approach, termed as MosaicKD, that allows us to take advantage of OOD data to conduct KD. Our motivation stems from the fact that, even though data from different domains exhibit divergent global distributions, their local distributions, such as patches in images, may however resemble each other. This observation further inspires us to leverage the local patterns, shared by the OOD and target-domain data, to resolve the domain shift problem in OOD-KD. As such, the core idea of MosaicKD is to synthesize in-domain data, of which the local patterns imitate those from real-world OOD data, while the global distribution, assembled from local ones, is expected to fool the pre-trained teacher. As shown in Figure 1, the shared local patterns are extracted from OOD data and re-assembled into in-domain data. Intuitively, this process is analogous to mosaic tiling, where tesserae are utilized to compose the whole art piece.

Specifically, in MosaicKD, we frame OOD-KD problem as a novel four-player min-max game involving a generator, a discriminator, a student, and a teacher, among which the former three are to be learned while the last one is pre-trained and hence fixed. The generator, as those in prior GANs, takes as input a random noise vector and learns to mosaic synthetic in-domain samples with locally-authentic and globally-legitimate distributions, under the supervisions back-propagated from the other three players. The discriminator, on the other hand, learns

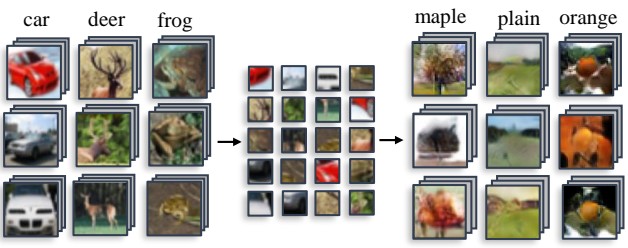

Out-of-Domain Data    Shared Local Patterns    In-Domain Mosaic

Figure 1: Natural images share common local patterns. In MosaicKD, these local patterns are first dissembled from OOD data and then assembled to synthesize in-domain data, making OOD-KD feasible.

to distinguish local patches extracted from the real-world OOD data and from the synthetic samples. The entire synthetic images are fed to both the pre-trained teacher and the to-be-trained student, based on which the teacher provides category knowledge for data synthesis and the student mimics the behavior of the teacher so as to carry out KD. The four players collaboratively reinforce one another in an adversarial fashion, and collectively accomplish the student training.

In short, our contribution is the first dedicated attempt towards OOD-KD, a highly practical yet largely overlooked problem, achieved through a novel scheme that mosaics in-domain data. The synthetic samples, generated via a four-player min-max game, enjoy realistic local structures and sensible global semantics, laying the ground for a dependable knowledge distillation from the pre-trained teacher. We conduct experiments over classification and semantic segmentation tasks across various benchmarks, and demonstrate that MosaicKD yields truly encouraging results much superior to those derived by its state-of-the-art competitors on OOD data.

## 2 Related Work

**Knowledge distillation.** Knowledge Distillation aims to craft a lightweight student model from cumbersome teachers [18], by either transferring network outputs [4, 18] or intermediate representations [41, 38]. In the literature, knowledge distillation and its variants largely rely on the premise that

original training data is available during distillation, which is vulnerable in real-world applications due to privacy or copyright reasons [33, 58, 44, 57, 45, 21, 43]. Recently, data-free knowledge distillation [30, 33, 10, 61, 59] has attracted attention from various research communities, which trains student model only with synthetic data. However, due to the difficulty in data synthesis without real-world samples, data-free KD usually leads to a degraded student on complicated tasks [61, 5]. Another solution to relax the conventional prerequisite on training data is to use some OOD samples. Unfortunately, it is found that naive knowledge distillation on OOD usually fails to learn a comparable student model from teachers [32, 27].

**Domain adaptation and generalization.** Most learning algorithms strongly rely on the premise that the source data for training and the target data for testing are independent and identically distributed [50], ignoring the OOD problem that is frequently encountered in real-world applications. In the literature, the OOD problem is usually addressed by domain generalization (DG) or adaptation (DA) [3, 64]. Adaptation is a popular technique for aligning the source and target domain [42, 11, 48], which typically requires the target domain to be accessible during training. In recent years, Domain adaptation has been extended to open-set settings where the label space of training and testing data are different [37]. In comparison, domain generalization is similar to domain adaptation, but does not requires the information from the target domain [3]. Domain generalization trains a model on the source domain only once and directly applied the model on the target domain [28, 12, 51, 16]. Despite the success of DA and DG in supervised learning, OOD problem is still under-studied in the context of knowledge distillation.

**Generative adversarial networks (GAN).** Generative adversarial network is initially introduced by Goodfellow *et al*. for image generation [13], where a generator is trained to fool a discriminator in an adversarial minimax game. In recent years, several works have been proposed to improve the performance of GANs, from the perspective of image quality [22, 23], data diversity [31, 56], and training stability [1]. Besides, Some explorations have been taken to make GAN training more efficient with limited data via transfer learning [52] or augmentation [62]. In this work, we study data synthesis in OOD settings, where the original training data is unavailable and thus conventional GAN technique can not be diretly deployed.

## 3 Out-of-Domain Knowledge Distillation

Without loss of generality, we study OOD problem in the context of image classification task. The domain underlying a dataset is defined as a triplet $\mathcal{D} = \{\mathcal{X}, \mathcal{Y}, P_{\mathcal{X} \times \mathcal{Y}}\}$, consisting of input space $\mathcal{X} \subset \mathbb{R}^{c \times h \times w}$, label space $\mathcal{Y} = \{1, 2, ..., K\}$ and joint distribution $P_{\mathcal{X} \times \mathcal{Y}}$ over $\mathcal{X} \times \mathcal{Y}$. Given a teacher model $T(x; \theta_t)$ optimized on target domain $\mathcal{D}$, vanilla KD trains a lightweight student model to imitate teacher's behaviour, by directly minimizing the empirical risk on the original domain:

$$\theta_s^* = \arg\min_{\theta_s} \mathbb{E}_{(x,y) \sim P_{\mathcal{X} \times \mathcal{Y}}} \left[ \ell_{\text{KL}}(T(x; \theta_t) \| S(x; \theta_s)) + \ell_{\text{CE}}(S(x; \theta_s), y) \right], \tag{1}$$

where $\ell_{\text{KL}}$ and $\ell_{\text{CE}}$ refers to the KL divergence and the cross entropy loss. However, when the original training domain $\mathcal{D}$ is unavailable and some alternative data from another domain $\mathcal{D}' = \{\mathcal{X}', \mathcal{Y}', P_{\mathcal{X} \times \mathcal{Y}}\}$ is used for training, Equation 1 may be problematic if the domain gap is huge. In this work, we focus on the out-of-domain problem in knowledge distillation, described as follows:

**Problem Definition (OOD-KD).** Given a teacher model $T(x; \theta_t)$ obtained from training domain $\mathcal{D} = \{\mathcal{X}, \mathcal{Y}, P_{\mathcal{X} \times \mathcal{Y}}\}$, the goal of OOD-KD is to craft a student model $S(x; \theta_s)$ only leveraging out-of-domain data from $\mathcal{D}' = \{\mathcal{X}', \mathcal{Y}', P_{\mathcal{X}' \times \mathcal{Y}'}\}$, where $\mathcal{X}' \neq \mathcal{X}$ and $\mathcal{Y}' \neq \mathcal{Y}$.

In OOD-KD, due to the domain divergence between OOD data and original training data, some important patterns may be the missing and the corresponding knowledge on these patterns might not be appropriately transferred from teachers to students. To address the OOD problem, we propose a novel assembling-by-dismantling approach to craft in-domain samples from out-of-domain ones, which effectively alleviates the domain gap between the transfer set and unavailable training set, making KD applicable on out-of-domain data.

## 4 Proposed Method

In the absence of original training data $X$, directly minimizing the risk on an OOD set $X'$ would be problematic due to the diverged data domain. In this work, we introduce a generative method for

OOD-KD, dubbed as MosaicKD, where a generator $G(z; \theta_g)$ is trained to synthesize a more helpful distribution $P_G$ for student learning. Specifically, MosaicKD is developed upon the distributionally robust optimization (DRO) framework which has been widely used to tackle domain shift [40, 16, 8]. Given a pre-defined distance metric $d(\cdot, \cdot)$ for distributions, the basic form of DRO framework can be formalized as the following:

$$\min_S \max_G \{\mathbb{E}_{x \sim P_G} [\ell_{\text{KL}}(T(G(z)) \| S(G(z)))] : d(P_G, P_{X'}) \leq \epsilon\} \tag{2}$$

In equation 2, $\ell_{\text{KL}}$ denotes the KL divergence for student learning, and $d(P_G, P_{X'})$ denotes the distribution distance between the generated samples and OOD data. The hyper-parameter $\epsilon$ specifies the radius of a ball space centered at $P_{X'}$. According to this definition, DRO framework aims to find the worst-case distribution from the searching space, which establishes an upper bound for the empirical risk of other distributions covered by the searching space. Ideally, if the target distribution $P_X$ of original training data exactly lies in the searching space, its empirical risk can effectively be optimized by the DRO framework. However, we would like to argue that this premise may be problematic in OOD settings, where $\mathcal{X} \neq \mathcal{X}'$ and $\mathcal{Y} \neq \mathcal{Y}'$. Note that if two distributions $P_{X_1}$ and $P_{X_2}$ are close in the input space under certain metric $d(\cdot; \cdot)$, their label space $\mathcal{Y}_1$ and $\mathcal{Y}_2$ should be also similar [49]. Based on this, distributions within the small $\epsilon$ ball space, centered at OOD distribution $P_{X'}$, are likely to share the same label space, i.e., $\mathcal{X} \approx \mathcal{X}'$ and $\mathcal{Y} \approx \mathcal{Y}'$, which obviously conflicts with the OOD settings. To this end, the target domain of original training data might not be covered by the search space and can not be bounded by the DRO framework. A remedy for this issue is to use a sufficiently large radius $\epsilon$. Unfortunately, it will only lead to intractable searching space, flooded with meaningless distributions.

## 4.1 Mosaicking to Distill

As mentioned above, the searching space built upon OOD data is insufficient for establishing an reliable upper bound for optimization. To address this problem, MosaicKD introduces a new way to construct the searching space based on local patches. Our motivation stems from the fact that, patterns of natural images are often organized hierarchically, where high-level patterns are assembled from low-level ones. Although the domain of original training data $X$ and OOD data $X'$ are diverged, their local patterns may still resemble each other. For example, the patterns of "fur" can be shared by different animal species from varied domains. Note that each images is assembled from local patches, we propose an assembling-by-dismantling strategy to re-organizes shared local patches and synthesize in-domain data for training.

**Patch Learning.** The first step towards MosaicKD is to extract local patterns from OOD data $X'$ and estimate the patch distribution for generation. Given an OOD dataset $X' = \{x'_1, x'_2, ..., x'_N; x'_i \in \mathbb{R}^{H \times W \times 3}\}$, we obtain patches through $L \times L$ cropping, which produces a patch dataset $C = \{c_1, c_2, ..., c_M; c_i \in \mathbb{R}^{L \times L \times 3}\}$. The patch size $L$ is an important hyper-parameter for MosaicKD. For example, if $L = W = H$, each patch will cover a full image, which contains all high-level features of original images. When we decrease the patch size to $L = 1$, then each patch only contains low-level color information. Obviously, small size $L$ can lead to more general patterns than large one, which are more likely to be shared by different domains. Besides, increasing the patch size will introduce more structural information, making the distribution of patches closer to the that of full images. In this work, we model local patch learning as a generative problem, where a generator $G(z; \theta_g)$ is trained to approximate the patch distribution by fooling a discriminator network $D(x; \theta_d)$. Note that our goal is to synthesize full images instead of pieces of patches, we train the generator $G(x; \theta_g)$ to produce images of full resolutions and craft patches on the fly. Let $C(\cdot)$ refers to the cropping operation, the objective of patch learning can be formulated as follows:

$$\min_G \max_D \mathcal{L}_{local}(G, D) = \mathbb{E}_{x' \sim P_{X'}} [\log D(C(x'))] + \mathbb{E}_{z \sim P_z} [\log(1 - D(C(G(z))))] \tag{3}$$

where $P_{X'}$ refers to the distribution of OOD data and $P_z$ refers to the prior distribution of the latent variable $z$. $C(x')$ and $C(G(z))$ refers to the cropped patches from OOD data and generated data. The main difference between Eqn. (3) and the objectives in vanilla GANs [13] lies in the patch-level discrimination, where MosaicKD only regularizes local patterns to be authentic, leaving the global structure unrestricted. As mentioned above, global patterns can be assembled from local ones, MosaicKD assembles these patches to synthesize in-domain data through label space aligning.

**Label Space Aligning.** As no inter-patch restriction is introduced in Eqn. (3), the generator may only produce images with meaningless global semantic, although their local patterns are plausible. In

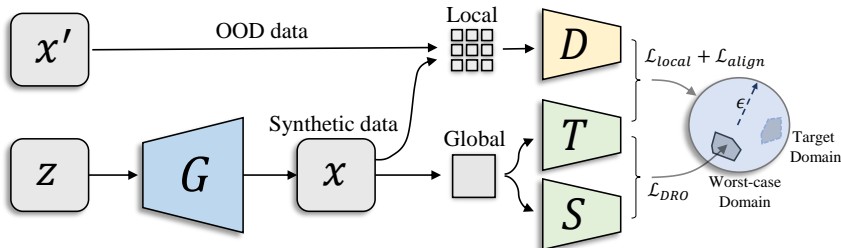

Figure 2: The framework of MosaicKD. We leverage the local patterns of OOD data and the category knowledge of teacher to synthesize locally-authentic and globally-legitimate samples for KD.

this step, we turn to the teacher model for more information for in-domain data synthesis. In KD, the teacher model is trained on the original training data $X$, whose output is a conditional probability $T(x; \theta_t) = p(y|x, \theta_t)$, which corresponds to the confidence that $x$ belongs to $y$-th category. To align the label space, a naive method is to maximize the confidence of teacher predictions, i.e., minimizing the entropy term $H(p(y|x, \theta_t))$. However, prior works have shown that such a simple probability maximization may only lead to some "rubbish samples" [14] without too much useful visual information for student training. To address this problem, we propose a regularized objective to align label space , which is formalized as:

$$\min_G \mathcal{L}_{align}(G, D, T) = \mathbb{E}_{z \sim P_z} \left[ \log(1 - D(C(G(z)))) + H\left[ p(y|G(z), \theta_t) \right] \right] \quad (4)$$

In Equation 4, the first term refers to the discrimination loss as mentioned in Equation 3, which regularizes the local patterns to be authentic. The second term refers to the entropy loss for confidence maximization, which works on full images and assembles local patterns to synethsize desired categories. This objective will be simultaneously optimized with Equation (3) to keep the authenticity of local patches.

**DRO in MosaicKD.** As aforementioned, Equation 3 regularizes the local patterns to be authentic and Equation 4 aligns the label space of synthetic data to that of training data. They collaboratively construct a new searching space for DRO framework as follows:

$$\min_S \max_G \mathcal{L}_{DRO}(G, D, S, T) = \left\{ \mathbb{E}_{z \sim p_z(z)} \left[ \ell_{\mathrm{KL}}(T(G(z)) \| S(G(z))) \right] : \mathcal{R}(G, D, T)) \leq \epsilon \right\} \quad (5)$$

where $\mathcal{R}(G, D, T)$ is a regularization term derived from Equation 3 and 4. Note that optimizing a generative adversarial networks as Equation 3 is equivalent to minimizing the Jensen–Shannon divergence of two patch distribution, i.e., $\ell_{\mathrm{JSD}}(P_{X'}^{patch}, P_G^{patch})$, the above regularization can be written as:

$$\mathcal{R}(G, D, T) = \ell_{\mathrm{JSD}}(P_{X'}^{patch}, P_G^{patch}) + \mathbb{E}_{z \sim P_z} \left[ H(p(y|G(z), \theta_t)) \right] \quad (6)$$

Regularization $\mathcal{R}(G, D, T)$ force the generator to leverage local patterns of OOD data for data synthesis, which leads to a special searching space defined on all possible schemes of patch assembling. Different to the conventional DRO, MosaicKD uses a small radius for robust optimization, where the target domain can be covered by the searching space. We relax the regularization of Equation 5 to obtain an optimizable DRO objective for training, formalized as follows:

$$\min_S \max_G \mathcal{L}_{DRO}(G, D, S, T) = \mathbb{E}_{x \sim P_z} \left[ \ell_{\mathrm{KL}}(T(G(z)) \| S(G(z))) - \lambda \mathcal{R}(G, D, T) \right) \quad (7)$$

## 4.2 Optimization

**Patch Discriminator.** For training efficiency, the discriminator in equation 3 can be implemented as a Patch GAN [19] with carefully designed receptive fields and patch overlap. Specifically, we stack several convolutional layers to build a fully convolutional network, whose output is a score map instead of a single true-or-fake scalar. Each score unit accepts a $L \times L$ local patches for discrimination. We apply an additional stride downsampling with step size $s$ on the score map to control the overlap between patches. A large step size $s$ will lead to more independent patches, which effectively reduce the structure restrictions in OOD images.

**Full Algorithm.** The full algorithm of MosaicKD is summarized in Alg. 1, where a generator $G(z; \theta_g)$, a discriminator $D(x; \theta_d)$, a fixed teacher model $T(x; \theta_t)$ and a student $S(x; \theta_s)$ are collectively optimized under the guidance of $\mathcal{L}_{local}$, $\mathcal{L}_{align}$ and $\mathcal{L}_{DRO}$.

---

**Algorithm 1** MosaicKD for out-of-domain knowledge distillation

---

**Input:** Pretrained teacher $T(x; \theta_t)$, student $S(x; \theta_s)$ and out-of-domain data $X'$.
**Output:** An optimized student $S(x; \theta_s)$

---

1: Initialize a generator $G(z; \theta_g)$ and a discriminator $D(z; \theta_d)$
2: **repeat**
3:      ▷ Patch Discrimination
4:      Sample a mini-batch of OOD data $x'$ from $X'$ and synthetic data $x$ from $G(z)$;
5:      update discriminator to distinguish fake patches from real ones using $\mathcal{L}_{local}$ from Eqn. 3;
6:      ▷ Generation
7:      Sample a mini-batch of generated data $x$ from $G(z)$;
8:      Update generator $G$ to:
9:          (a) fool the discriminator $D$ using $\mathcal{L}_{local}$ from Eqn. 3;
10:         (b) align label space with teacher $T$ using $\mathcal{L}_{align}$ from Eqn. 4;
11:         (c) fool the student $S$ using $\mathcal{L}_{DRO}$ from Eqn. 7;
12:      ▷ Knowledge Distillation
13:      **for** $j$ steps **do**:
14:         Sample generated samples from $G(z)$;
15:         Update student through knowledge distillation using $\mathcal{L}_{DRO}$ from Eqn. 7
16:      **end for**
17: **until** converge

---

## 5 Experiments

### 5.1 Settings

**Datasets.** The proposed method is evaluated on two mainstream vision tasks, *i.e.*, image classification and semantic segmentation. Four datasets are considered in our experiments as in-domain training set, including CIFAR-100 [26], CUB-200 [53], Stanford Dogs [24] and NYUv2 [34]. For OOD-KD, we substitute original training data with OOD data, including CIFAR-10 [26], Places365 [63], ImageNet [9] and SVHN [36].

**Evaluation metrics.** For image classification, accuracy and Frechet Inception Distance (FID) are used to evaluate different methods. FID indicates the divergence of two datasets, which was originally used to assess the synthesis quality of GANs [17]:

$$FID = \|\mu_1 - \mu_2\|_2^2 + tr\left(\Sigma_1 + \Sigma_2 - 2\left(\Sigma_1\Sigma_2\right)^{1/2}\right) \tag{8}$$

where $(\mu_1, \Sigma_1)$ and $(\mu_2, \Sigma_2)$ are the mean and covariance statistics of generated and original samples. For semantic segmentation, we use mean Intersection of Unions (mIoU) as the performance metric. More details about datasets, training protocol, and metrics can be found in supplementary materials.

### 5.2 Results of Knowledge Distillation

**CIFAR-100.** Table 1 reports the results of knowledge distillation on CIFAR-100 dataset. Here we use CIFAR-10, ImageNet, Places365 and SVHN as OOD data to evaluate MosaicKD for OOD settings. We compare the proposed MosaicKD to various baselines, including data-free KD methods (DAFL [7], ZSKT [33], DeepInv. [61], DFQ [8]) and OOD-KD methods naively adapted from state-of-the-art KD approaches (BKD [18], Balanced [35], FitNet [41], RKD [38], CRD [47] and SSKD [54]).

As shonw in Table 1, despite the mismatched distributions, conventional KD approaches still learn some useful but incomplete knowledge from OOD data (*i.e.*, yielding significantly superior performance to random guessing), which indicates the existence of shared patterns between OOD data and training data. Further, some exploration was taken to evaluate importance of category balance and representation transfer for OOD-KD. First, we balance the OOD data by re-sampling the scarce categories according to teacher's predictions. However, results show that balancing the OOD data may not help students learn correct class information, because most samples in OOD data are just misclassified outliers. In the context of OOD settings, the balance operation may lead to over-fitting on outliers, which may further degrade the student performance on the test set. As mentioned before,

| Method | Data | resnet-34 resnet-18 | vgg-11 resnet-18 | wrn40-2 wrn16-1 | wrn40-2 wrn40-1 | wrn40-2 wrn16-2 | Average |
|---|---|---|---|---|---|---|---|
| Teacher | CIFAR-100 (Original Data) | 78.05 | 71.32 | 75.83 | 75.83 | 75.83 | 75.37 |
| Student | | 77.10 | 77.10 | 65.31 | 72.19 | 73.56 | 73.05 |
| KD [18] | | 77.87 | 75.07 | 64.06 | 68.58 | 70.79 | 72.27 |
| DAFL [7] | Data-Free | 74.47 | 54.16 | 20.88 | 42.83 | 43.70 | 47.20 |
| ZSKT [33] | | 67.74 | 54.31 | 36.66 | 53.60 | 54.59 | 53.38 |
| DeepInv. [61] | | 61.32 | 54.13 | 53.77 | 61.33 | 61.34 | 58.38 |
| DFQ [8] | | 77.01 | 66.21 | 51.27 | 54.43 | 64.79 | 62.74 |
| KD [18] | CIFAR-10 (OOD Data) | 73.55 | 68.04 | 47.47 | 61.17 | 63.48 | 62.74 |
| Balanced [35] | | 68.54 | 64.14 | 50.50 | 56.50 | 57.33 | 59.40 |
| FitNet [41] | | 70.14 | 67.52 | 50.31 | 60.17 | 60.60 | 63.15 |
| RKD [38] | | 67.45 | 63.06 | 45.37 | 53.29 | 57.10 | 57.25 |
| CRD [47] | | 71.23 | 66.48 | 47.00 | 59.59 | 61.37 | 61.13 |
| SSKD [54] | | 73.81 | 68.72 | 49.57 | 60.71 | 64.61 | 63.48 |
| Ours | | **77.01** | **71.56** | **61.01** | **69.14** | **69.41** | **69.55** |
| KD [18] | ImageNet[†] (OOD Subset) | 50.89 | 50.52 | 36.54 | 36.87 | 41.69 | 43.30 |
| Balanced [35] | | 41.74 | 47.04 | 31.61 | 29.57 | 35.65 | 37.12 |
| FitNet [41] | | 60.15 | 58.23 | 42.63 | 44.21 | 48.53 | 50.75 |
| RKD [38] | | 40.26 | 35.80 | 31.15 | 24.95 | 34.48 | 33.32 |
| Ours | | **75.81** | **68.94** | **59.32** | **66.61** | **67.36** | **67.60** |
| KD [18] | Places365[†] (OOD Subset) | 43.49 | 46.24 | 33.28 | 31.39 | 36.37 | 38.15 |
| Balanced [35] | | 28.16 | 38.85 | 23.22 | 21.54 | 28.62 | 28.08 |
| FitNet [41] | | 54.08 | 54.15 | 36.33 | 44.21 | 38.74 | 45.50 |
| RKD [47] | | 30.25 | 33.06 | 28.07 | 21.12 | 21.12 | 26.72 |
| Ours | | **74.70** | **68.55** | **56.70** | **65.34** | **65.89** | **66.23** |
| KD [18] | SVHN (OOD Data) | 31.55 | 34.00 | 19.77 | 23.07 | 24.75 | 26.63 |
| Balanced [35] | | 26.93 | 29.34 | 16.18 | 18.96 | 21.50 | 22.58 |
| FitNet [41] | | 33.69 | 36.22 | 20.02 | 23.72 | 25.41 | 27.81 |
| RKD [38] | | 26.83 | 27.31 | 18.09 | 22.55 | 24.29 | 23.81 |
| Ours | | **47.18** | **37.63** | **31.87** | **45.84** | **44.40** | **41.38** |

Table 1: Test accuracy (%) of student networks trained with the following settings: conventional KD with original training data, data-free KD with synthetic data, and OOD-KD with OOD data. †: As Places365 and ImageNet contain some in-domain samples, we craft OOD subsets with low teacher confidence (high entropy) from the original dataset, so as to match our OOD setting.

| Method | Data | FLOPs | mIoU |
|---|---|---|---|
| Teacher | NYUv2 | 41G | 0.519 |
| Student | | 5.54G | 0.375 |
| ZSKT | Data-Free | 5.54G | 0.364 |
| DAFL | | 5.54G | 0.105 |
| KD | ImageNet | 5.54G | 0.406 |
| Ours | | 5.54G | **0.454** |

Table 2: Mean Intersection over Union (mIoU) of student models on NYUv2 data set.

| Method | CUB-200 | Stanford Dogs |
|---|---|---|
| Teacher | 49.41 | 56.65 |
| Student | 41.44 | 48.61 |
| KD | 11.07 | 10.24 |
| Balanced | 4.56 | 6.42 |
| FitNet | 18.12 | 19.13 |
| Ours | **26.11** | **28.02** |

Table 3: Test accuracy of student networks on fine-grained datasets.

OOD data and original data may share some local patterns, which can be extracted by shallow layers of networks. We apply four representation transfer approaches, i.e., FitNet, RKD, CRD and SSKD to study the their role in OOD-KD. Compared with RKD that focus instance relation, we found that response-based methods like fitnet can transfer more helpful information in OOD settings, where the student directly imitates the teacher's intermediate outputs of teachers. In general, transferring low-level features sometimes can be helpful for OOD-KD. However, note that CRD works on the high-level representation extracted from the penultimate layer, transferring these knowledge may be inappropriate for OOD-KD because high-level features may be unrelated to target tasks.

In this work, we handles the OOD-KD problem as a generative problem, instead of directly using OOD data for training. The proposed method leverage local patterns of OOD data for data synthesis, where some task-related patterns will be "assembled" from shared local patches. Results show that these re-assembled data can effectively transfer knowledge from teachers to students. In Table 1, we also extend our method to different types of OOD sets. We found that the performance of MosaicKD

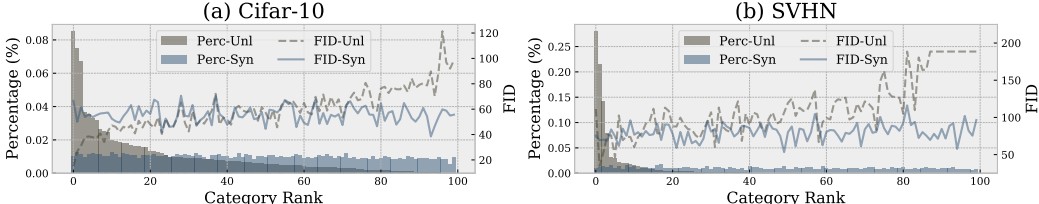

Figure 3: Statistical information of OOD data and out generated data. Category percentage (the first y-axis) and FID score (the second y-axis) to original data (CIFAR-100) is reported.

| Patch size | | 1 | 2 | 4 | 8 | 18 | 22 | 32 |
|---|---|---|---|---|---|---|---|---|
| CIFAR-10 | Acc. | 52.22 | 55.58 | 58.94 | 61.01 | **61.03** | 58.94 | 51.34 |
| | FID | 2.43 | 4.47 | 8.78 | 16.82 | 22.81 | 26.07 | 28.30 |
| Places365 | Acc. | 46.44 | 45.90 | 50.67 | **56.70** | 53.99 | 53.44 | 40.29 |
| | FID | 12.32 | 14.77 | 21.32 | 30.09 | 35.54 | 38.64 | 41.41 |
| SVHN | Acc. | 19.83 | 21.86 | **32.09** | 31.87 | 21.05 | 22.08 | 20.54 |
| | FID | 93.39 | 146.76 | 145.64 | 143.72 | 148.33 | 147.25 | 148.94 |

Table 4: Test accuracy (%) of students obtained with different patch sizes. The Patch FID score between OOD data and original data is also reported. Results show that our approach requires smaller patch sizes to handle severe domain discrepancies.

is related to the degree of domain divergence between OOD data and original data. For example, ImageNet is an object recognition dataset while Places365 is a scene classification dataset. Results show that, for the target data CIFAR-100, MosaicKD can achieve better performance on ImageNet compared to Places365.

**Fine-grained Classification.** To further study the effectiveness of our approach, we conduct knowledge distillation on fine-grained datasets as shown in Table 3, using Places365 as OOD data. OOD-KD on fine-grained data is a challenging problem, because different categories is visually similar. Results show that our method achieves superior performance compared to baselines methods.

**Semantic Segmentation.** Semantic segmentation can also be viewed as a classification task, where the network is trained to predict the category of each pixel. We apply our method to the NYUv2 dataset, following the protocol in [10]. The teacher network is a deeplab v3 network with resnet-50 backbone. The student is a freshly initialized deeplabv3-MobileNetv2 model. Our method can effectively improve the knowledge transfer on OOD data and achieve competitive results even compared to vanilla KD settings.

## 5.3 Quantitative Analysis

**Data balance and FID.** Figure 3 provides some statistical information of OOD data and generated samples, including the category balance predicted by teachers and the per-class FID scores. The category is ranked according to their percentages. Note that the original CIFAR-10 dataset only contains 10 categories, which is very limited compared with the 100 categories of CIFAR-100. As illustrated in Figure 3 (a), some CIFAR-100 categories are missing in CIFAR-10. Besides, the large FID between OOD data and original training data also indicates that, even though some samples are categorized to some classes by the teacher, they may still belong to outliers. by contrast, our method successfully balances different CIFAR-100 categories and alleviates the domain gap (lower class FID), especially for unbalanced categories.

**The influence of patch size.** Patch size plays an essential role in our method, which determines the flexibility of data synthesis. The optimal patch size actually depends on the divergence between OOD data and in-domain data. For OOD data with large domain discrepancy, small patches are usually required due to the limited local similarity. As shown in Table 4, we evaluate our method with different patch sizes and report the test accuracy of student models as well as the patch FID. According to the Table 4, we find that for OOD data like CIFAR-10 and Places365, a large patch size (e.g., 18) can be used for student learning. However, for SVHN dataset, a smaller patch size would be more appropriate, as SVHN severely diverges from CIFAR-100.

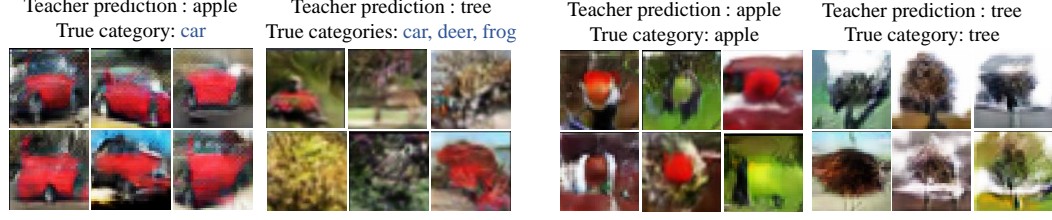

| Teacher prediction : apple | Teacher prediction : tree | Teacher prediction : apple | Teacher prediction : tree |
| True category: car | True categories: car, deer, frog | True category: apple | True category: tree |

(a) W/O Patch Learning           (b) Patch Learning (Ours)

Figure 4: Visualization of synthetic data with and without patch learning. GANs without patch learning will be trapped by OOD data and fails to present correct semantic for different categories (highlighted in blue). In our method, the semantic can be correctly aligned with target domain.

| Method | wrn40-2 wrn16-1 | wrn40-2 wrn40-1 | wrn40-2 wrn16-2 |
|---|---|---|---|
| ours | **61.01** | **69.14** | **69.41** |
| w/o patch | 51.34 | 56.23 | 57.23 |
| w/o disc. | 44.05 | 58.11 | 59.73 |
| w/o adv. | 55.11 | 67.57 | 68.25 |

Table 5: Test accuracy (%) of students in different ablation settings.

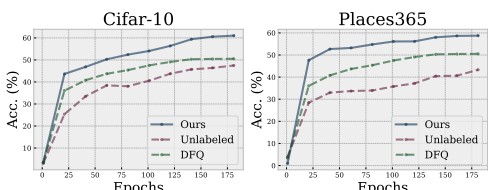

Figure 5: Training efficiency of our method compared with DFQ and vanilla KD.

**Ablation study.** In this section, we conduct ablation to understand the role of patch learning further. We consider the following settings: (a) MosaicKD (b) mosaicKD without patch learning (c) mosaicKD without discriminator (d) mosaicKD wihout adversarial training. As shown in Table 5, we find that full image discrimination sometimes even lead to worse results than MosaicKD without discrimination. Figure 4 visualizes the synthetic data with or without patch learning. Results show that the generator without patch learning is trapped by the label space of OOD data, failing to synthesize in-domain categories like trees and apples.

# 6 Discussion

**Relation to unlabeled knowledge distillation.** In the literature of knowledge distillation, a slice of works also use unlabeled data for student learning. However, they either make an i.i.d. assumption about unlabeled data and original training [60], or assume that there are sufficient in-domain samples inside unlabeled set [55]. In this work, we allow the unlabeled set to be fully OOD, which is more challenging than existing unlabeled settings.

**Relation to data-free knowledge distillation.** The commonality between the data-free algorithm and MosaicKD lies in that they both solve the KD problem through data synthesis. However, data-free KD leverages some simple priors such as category confidence [7] and gaussian assumptions [61], which ignores the structural details in natural images. By contrast, MosaicKD achieves data synthesis in an assembling-by-dismantling manner, where natural patterns can be utilized to improve synthesis quality. The training curves of different methods can be found in Figure 5.

# 7 Conclusion

In this work, we propose a novel approach, termed as MosaicKD, that enables us to take advantage of only the OOD data for KD. MosaicKD follows a assembling-by-dismantling scheme where a synthetic sample is generated by locally resembling the real-world OOD data while globally fooling the pre-trained teacher. This is technically achieved through a handy yet effective four-player min-max game, in which a generator, a discriminator, and a student network are learned in the presence of a pre-trained teacher. We validate MosaicKD over classification and semantic segmentation tasks across various benchmarks, and showcase that it yields results significantly superior to the state-of-the-art techniques on OOD data.

# 8 Acknowledgements and Disclosure of Funding

This work is supported by National Natural Science Foundation of China (U20B206661976186), Key Research and Development Program of Zhejiang Province (2020C01023), the Major Scientific Research Project of Zhejiang Lab (No. 2019KD0AC01), the Fundamental Research Funds for the Central Universities, Alibaba-Zhejiang University Joint Research Institute of Frontier Technologies, Start-Up Grant from National University of Singapore (R-263-000-E95-133), and MOE AcRF TIER 1 FRC Research Grant (R-263-000-F14-114).

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
