# Mosaicking to Distill:
# Knowledge Distillation from Out-of-Domain Data
# – *Supplementary Material* –

**Gongfan Fang**[1,4], **Yifan Bao**[1], **Jie Song**[1], **Xinchao Wang**[2], **Donglin Xie**[1]
**Chengchao Shen**[3], **Mingli Song**[1]
[1]Zhejiang University, [2]National University of Singapore, [3]Central South University
[4]Alibaba-Zhejiang University Joint Institute of Frontier Technologies
{fgf,yifanbao,sjie,donglinxie,brooksong}@zju.edu.cn
xinchao@nus.edu.sg, scc.cs@csu.edu.cn

In this document, we provide details and supplementary materials that cannot fit into the main manuscript due to the page limit. Specifically, we provide optimization details of MosaicKD in Sec. A, experimental settings in Sec. B, and more experimental results in Sec. C.

## A   Optimization Details

### A.1   Alleviating Mode Collapse.

In this work, we deploy a generator to synthesize the transfer set for knowledge distillation. Nevertheless, GANs are known to suffer from mode collapse and fail to produce diverse patterns. To this end, we leverage both OOD data and synthetic ones to train our student models, so that the generator does not need to synthesize all samples for KD. Besides, an additional balance loss is deployed to alleviate mode collapse during training, defined as:

$$L_{balance} = -H(\mathbb{E}_{x \sim P_G}(p(y|x, \theta_t)))  \tag{1}$$

where $p(y|x, \theta_t)$ is the probability prediction after softmax, and $P_G$ denotes the distribution of generated samples. Minimizing Eq. (1) will enforce the class to be balanced during the synthesizing process.

### A.2   Objectives of MosaicKD.

As shown in the main manuscript, MosaicKD aims to solve a distributionally robust optimization (DRO) problem as follows:

$$\min_{S} \max_{G} \{\mathbb{E}_{x \sim P_G}\left[\ell_{\text{KL}}(T(x; \theta_t) \| S(x; \theta_s))\right] : \mathcal{R}(G, D, T)) \leq \epsilon\}  \tag{2}$$

where $\mathcal{R}(G, D, T)) \leq \epsilon$ defines the search space, i.e., a ball space with radius $\epsilon$ centered at an distribution satisfying $\mathcal{R}(G, D, T)) = 0$. The specific form of center distribution is unknown, but we can still train a generator $G$ to approximate it. Note that Eq. (2) is intractable due to the non-differentiable condition on the search space. With the help of lagrange duality, we can re-express the inner part of Eq. (2) as follows:

$$
\begin{aligned}
&\max_{G} \{\mathbb{E}_{x \sim P_G}\left[\ell_{\text{KL}}(T(x; \theta_t) \| S(x; \theta_s))\right] : \mathcal{R}(G, D, T)) \leq \epsilon\} \\
&= \max_{G} \min_{\lambda \geq 0} \{\mathbb{E}_{x \sim P_G}\left[\ell_{\text{KL}}(T(x; \theta_t) \| S(x; \theta_s))\right] + \lambda \cdot (\epsilon - \mathcal{R}(G, D, T)))\} \\
&\leq \min_{\lambda \geq 0} \max_{G} \{\lambda \epsilon + \mathbb{E}_{x \sim P_G}\left[\ell_{\text{KL}}(T(x; \theta_t) \| S(x; \theta_s))\right] - \lambda \cdot \mathcal{R}(G, D, T)))\} \\
&= \min_{\lambda \geq 0} \{\lambda \epsilon + \max_{G} \{\mathbb{E}_{x \sim P_G}\left[\ell_{\text{KL}}(T(x; \theta_t) \| S(x; \theta_s))\right] - \lambda \cdot \mathcal{R}(G, D, T)))\}\}
\end{aligned}
\tag{3}
$$

35th Conference on Neural Information Processing Systems (NeurIPS 2021).

where $\lambda$ is Lagrangian multiplier and $\lambda\epsilon$ is a constant term. If $\mathcal{R}(G, D, T)) \leq \epsilon$, we choose $\lambda = 0$, i.e., no restriction on $\mathcal{R}(G, D, T))$, to obtain the minimal cost. If $\mathcal{R}(G, D, T)) > \epsilon$, then a large $\lambda$ should be applied as a penalization. According to the derivation of Eq. (3), we obtain a relaxed version of the intractable Eq. (2), expressed as follows:

$$\min_{S} \max_{G} \mathcal{L}_{DRO}(G, D, S, T) = \mathbb{E}_{x \sim P_G}\left[\ell_{\text{KL}}(T(x; \theta_t), S(x; \theta_s))\right] - \lambda \mathcal{R}(G, D, T)) \qquad (4)$$

### A.3 GAN Training and JS Divergence.

Following the conventions of prior works, we write the GAN training objective as follows,

$$\min_{G} \max_{D} V(D, G) = \mathbb{E}_{x \sim P_{data}}\left[\log D(x)\right] + \mathbb{E}_{z \sim P_z}\left[log(1 - D(G(z)))\right]. \qquad (5)$$

As proposed in [1], for a fixed generated $G$ and a given data distribution $P_{data}$, the optimal discriminator $D$ is achieved when

$$D^*(x) = \frac{P_{data}(x)}{P_{data}(x) + P_G(x)} \qquad (6)$$

We then replace the discriminator in Eq. (5) with the optimal one $D^*$, which leads to the following optimization for generator $G$:

$$
\begin{aligned}
\min_{G} V(G, D^*) &= \mathbb{E}_{x \sim P_{data}}\left[\log D^*(x)\right] + \mathbb{E}_{z_z}\left[log(1 - D^*(G(z)))\right] \\
&= \mathbb{E}_{x \sim P_{data}}\left[\log D^*(x)\right] + \mathbb{E}_{x \sim P_G}\left[log(1 - D^*(x))\right] \\
&= \mathbb{E}_{x \sim P_{data}}\left[\log \frac{P_{data}(x)}{P_{data}(x) + P_G(x)}\right] + \mathbb{E}_{x \sim P_G}\left[log(\frac{P_G(x)}{P_{data}(x) + P_G(x)})\right] \quad (7) \\
&= -log(4) + \ell_{\text{KL}}(P_{data}\|\frac{P_{data} + P_G}{2}) + \ell_{\text{KL}}(P_G\|\frac{P_{data} + P_G}{2}) \\
&= -log(4) + 2 \cdot \ell_{\text{JSD}}(P_{data}\|P_G)
\end{aligned}
$$

Therefore, as mentioned in the manuscript, we optimize generative adversarial networks to minimize the regularization term $R(G, D, T)$, which is equivalent to optimizing the JS divergence between patch distributions.

## B  Experimental Settings

**Datasets.** The proposed method is evaluated on two mainstream vision tasks, *i.e.*, image classification and semantic segmentation, over four labeled datasets for teacher training and four OOD data for student learning, as summarized in Table 1. Note that CIFAR-100, ImageNet, and Places365 may contain in-domain categories. We craft OOD subset from the full ImageNet and Places365 datasets by selecting samples with low prediction confidence, as described in Algorithm B. These OOD subsets can be viewed as out-of-domain data for CIFAR-100. Besides, we resize the OOD data to the same resolution as in-domain data, e.g., $32 \times 32$ for CIFAR-100, $64 \times 64$ for fine-grained datasets, and $128 \times 128$ for NYUv2.

---

**Algorithm 1** OOD subset selection

**Input:** dataset $D$, Pretrained teacher $T(x; \theta_t)$,
**Output:** OOD subset $D'$

1: $H \leftarrow []$
2: **for** $x_i$ in D **do**:
3:     obtain prediction $p(y|x_i) = T(x)$
4:     calculate the entropy $h_i = H(p(y|x_i))$
5:     $H$.append($h_i$)
6: **end for**
7: index $\leftarrow$ topk-index($H$);
8: $D' \leftarrow D[\text{index}]$;
9: return $D'$

---

**Network Training.** In this work, all teacher models are trained using the in-domain datasets listed in Table 1 with cross entropy loss. We use SGD optimizer with $\{lr = 0.1, weight\_decay = 1e-4, momentum = 0.9\}$ and train each model for 200 epochs, with cosine annealing scheduler. In knowledge distillation, student models are crafted using unlabeled datasets, where only the soft targets from teachers are utilized. We use the same training protocols as the teacher training and report the best student accuracy on test sets. We use Adam for optimization, with hyper-parameters $\{lr = 1e-3, \beta_1 = 0.5, \beta_2 = 0.999\}$ for the generator and discriminator.

| In-Domain Data | Training | Testing | Num. Classes |
|---|---|---|---|
| CIFAR-100 | 50,000 | 10,000 | 100 |
| CUB200 | 5,994 | 5,794 | 200 |
| Stanford Dogs | 12,000 | 8,580 | 120 |
| NYUv2 | 795 | 654 | 13 |
| **OOD Data** | **Training** | **Testing** | **Num. Classes** |
| CIFAR-10 | 50,000 | 10,000 | 100 |
| ImageNet-OOD | 50,000 | - | - |
| Places365-OOD | 50,000 | - | - |
| SVHN | 73,257 | 26,032 | 10 |
| ImageNet | 1,281,167 | 50,000 | 1000 |
| Places365 | 1,803,460 | 36,500 | 365 |

Table 1: Statistical information of in-domain and out-of-domain datasets

| Input: $z \in \mathbb{R}^{100} \sim \mathcal{N}(0, I)$ |
|---|
| Linear(100) $\to 8 \times 8 \times 128$ |
| Reshape, BN, LeakyReLU |
| Upsample2$\times$ |
| $3 \times 3$ Conv128 $\to 128$, BN, LeakyReLU |
| Upsample2$\times$ |
| $3 \times 3$ Conv128 $\to 64$, BN, LeakyReLU |
| $3 \times 3$ Conv64 $\to 3$, Sigmoid |

Table 2: Generator archicture for CIFAR-100. We add more convolutional layers and upsample layers for datasets with larger resolution.

| Input: $x \in \mathbb{R}^{32 \times 32 \times 3}$ |
|---|
| $3 \times 3$ Conv3 $\to 64$, stride $= 2$ |
| BN, LeakyReLU |
| $3 \times 3$ Conv64 $\to 128$, stride $= 2$ |
| BN, LeakyReLU |
| $3 \times 3$ Conv128 $\to 1$, stride$^{\dagger} = 1$ |
| Sigmoid |

Table 3: Patch Discriminator archicture for CIFAR-100. †: The final stride controls the patch overlap of MosaicKD.

**Generator and Discriminator.** The architecture of GAN for CIFAR-100 dataset is illustrated in Tables 2 and 3. For CUB-200 ($64 \times 64$) and NYU ($128 \times 128$), we add more convolutional layers and upsampling or sampling layers to generate high-resolution images.

# C  More Experimental Results

## C.1  Patch Overlap

Given a fixed patch size, the overlap between patches plays an important role in patch learning. The overlap is controlled by interval sampling in the patch discriminator. Note that the discriminator produces a prediction map to predict each small region on the original image, which means that distant predictions should share less information. We add a prediction stride to the final discrimination to control the patch overlap. Table 4 shows the student accuracy obtained with different patch overlaps, where a larger stride corresponds to a smaller overlap. The results show that increasing stride does not benefit the students' accuracy. Note that we use the patch GAN architecture for patch learning, which contains internal stride operations within the discriminator. These stride operations already provide an appropriate overlap for patch learning. Besides, a larger stride also means fewer training samples, which may be harmful to the GAN training.

| Stride | wrn40-2 wrn16-1 | wrn40-2 wrn40-1 | wrn40-2 wrn16-2 |
|---|---|---|---|
| stride=1 | **61.01** | **69.14** | **69.41** |
| stride=2 | 59.56 | 60.26 | 63.46 |
| stride=3 | 42.35 | 54.32 | 57.36 |
| stride=4 | 46.07 | 55.12 | 54.82 |

Table 4: Influence of patch overlap. We control the patch overlap by using different strides at the prediction layer of the patch discriminator.

|  (a) No regularization | (b) Full-image regularization | (c) Patch regularization |

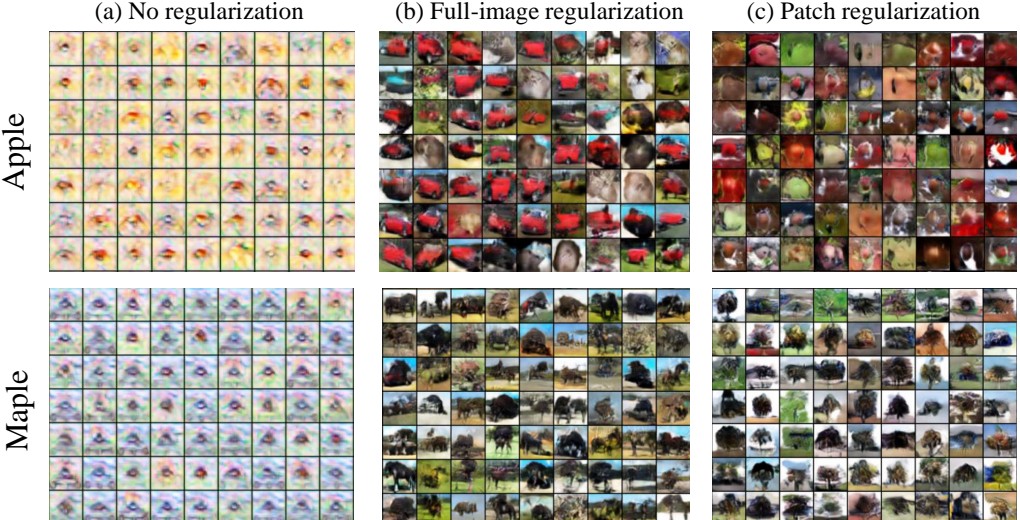

Figure 1: Synthetic images from the generator: (a) without regularization, (b) with full image regularization, and (c) with patch regularization.

### C.2  DRO Regularization

In MosaicKD, the search space is regularized by $\mathcal{L}_{local}$ and $\mathcal{L}_{align}$, which enforces the generated samples to be locally authentic and globally legitimate. We take a further study on the above regularization to show their significance for MosaicKD. As illustrated in 1, we visualize the generated samples with different regularizations. In Figure 1(a), no regularization is applied on the generator, and we naively maximize the teacher's confidence, which will lead to some inferior samples [2]. In Figure 1(b), the discriminator makes decisions on full images, and, to some extent, the generator will be trapped by the class semantic of OOD data, i.e., synthesizing a car-like apple or a horse-like maple. Figure(c) showcases the synthetic samples of MosaicKD, which reveals the correct semantic of task-related classes.

### C.3  ImageNet Results

Table 5 provides the student's accuracy on $32 \times 32$ ImageNet dataset with 1000 categories. We use Places365 [6] as the OOD data and resize all samples to $32 \times 32$ for training. Results show that our approach is indeed beneficial for the OOD-KD task.

| Method | Data | resnet-56 resnet-20 | resnet-56 mobilenetv2 |
|--------|------|---------------------|------------------------|
| Teacher | ImageNet (Original Data) | 41.28 | 41.28 |
| Student |  | 32.20 | 32.48 |
| KD [3] |  | 32.18 | 32.55 |
| KD [3] | Places365 (OOD Data) | 21.76 | 10.25 |
| Balanced [4] |  | 21.09 | 11.34 |
| FitNet [5] |  | 21.45 | 13.12 |
| Ours |  | **26.51** | **20.46** |

Table 5: Test accuracy (%) of student networks on ImageNet. We use the full places365 dataset as transfer set for OOD-KD.