# OpenReview forum: "Mosaicking to Distill: Knowledge Distillation from Out-of-Domain Data"
_NeurIPS.cc/2021/Conference — NeurIPS 2021 Poster_

### Official Review · Reviewer_aTEX · 2021-07-14

**Rating:** 8
**Confidence:** 4

**Summary:**

The authors propose a knowledge distillation algorithm for out-of-domain data, where local similarity between different domains is leveraged to transfer knowledge across models. In particular, the authors first present a basic method based on distributionally robust optimization and discuss the its problem in OOD settings. Then the authors improve the basic method by introducing a local loss and an aligning loss. The proposed method, named MosaicKD aims at synthesizing locally-authentic and globally-legitimate samples for KD. The authors demonstrate that MosaicKD can be applied to various OOD data and outperforms several data-free and data-driven methods (e.g. ZSKT, DFQ and CRD)

**Limitations And Societal Impact:**

I have some questions about the proposed method:
1) what is the relation between MosaicKD and existing data augmentation methods, like mixup, or perturbation. In the literature of KD, some works adopts augmented data for training [1,2]. Please make a clarification for their difference.
2) In table 4, a too small patch size leads to `best ` FID but worse student performance. Please provide a discussion about this phenomenon. In particular, what kind of information is missing in small patches which degrades student’s accuracy.
[1] Heo, B., Lee, M., Yun, S., & Choi, J. Y. (2019, July). Knowledge distillation with adversarial samples supporting decision boundary. In Proceedings of the AAAI Conference on Artificial Intelligence (Vol. 33, No. 01, pp. 3771-3778).
[2] Fu, J., Geng, X., Duan, Z., Zhuang, B., Yuan, X., Trischler, A., ... & Dong, H. (2020). Role-wise data augmentation for knowledge distillation. arXiv preprint arXiv:2004.08861.

**Main Review:**

pros:
It is the first attempt to study knowledge distillation with out-of-domain data. The author utilizes the local similarity to alleviate the domain gap between OOD data and source training data. The idea of crafting in-domain data from low-level patches is novel and practical. The author also demonstrate that patch learning can help the generator escape from the OOD distribution and synthesize correct semantic, as the global structure is freed by the patch-level training.

cons:
It is not clear to me how the authors determine the patch size (the receptive fields of Patch Discriminator) and the overlap between patches (the stride applied on discriminator’s logits) for different OOD data. According to the code in supplementary materials, the stride is set to 1 by default. Please provide more details about the design of patch discrimination. Besides, some equations and derivations are not clearly presented in the main paper, such as the Equation 7 and its derivations.

========= post rebuttal ===========
I have read the rebuttal. The rebuttal addresses all my concerns. I will change my score to 8.

**Time Spent Reviewing:**

4

---

> ### Author Response · Authors · 2021-08-10
> **Response to Reviewer aTEX**
>
> We sincerely thank the reviewer for the constructive comments on our work.
>
> **Q1: How the authors determine the patch size and the overlap between patches for different OOD data**
> **A1:** Thanks for the question. The patch size and overlap are important hyper-parameters for our method. In our experiments, we use a fixed 8x8 patch size for all 32x32 datasets based on the empirical results in Table 4 of the main paper; we set the discrimination stride to 1, as described in Table 4 of the supplemental materials.
>
> **Q2: What is the relation between MosaicKD and existing data augmentation methods, like mixup, or perturbation.**
> **A2:** We thank the reviewer for the question. Typically, augmentation methods are not designed to significantly change the data domain. For example, mixup makes a linear interpolation between images and produces mixed labels for training. The label space of mixed data is still trapped by the OOD domain. Perturbation adds some noise to images and does not change their real categories either. By contrast, our method adopts an assembling-by-dismantling strategy which is able to achieve cross-domain adaptation. As shown in Figure 4, synthetic images present semantics of the target domain, although they are crafted using OOD data.
>
> **Q3: What kind of information is missing in small patches which degrade student’s accuracy**
> **A3:** Let's assume that a 1x1 patch size is used in MosaicKD, where each patch is an RGB pixel. These patches only reveal the color distribution of OOD data, ignoring all relations between pixels. In this case, very limited information can be learned by the generator for mosaicking. Therefore, smaller patch sizes will lead to less structural patterns and larger searching space for DRO, which both degrade the student performance.

---

### Official Review · Reviewer_jgX3 · 2021-07-14

**Rating:** 7
**Confidence:** 4

**Summary:**

This paper studies knowledge distillation in out-of-domain settings and proposes a generative method called MosaicKD to handle the agnostic domain gap between OOD transfer set and original training data. The core idea of MosaicKD lies in the observation that different domain may still share some local patterns, which can be re-assembled for cross-domain synthesis. The author introduces a four-player adversarial training method to learn a locally-authentic and globally-legitimate distribution for knowledge distillation.

**Limitations And Societal Impact:**

Part of these have been discussed in the paper.

**Main Review:**

**[Strengths]:**

The motivation of this work is well presented and the idea of patch learning is novel. In my opinion, knowledge distillation with DRO is reasonable and practical for OOD settings as the goal is to optimize the risk on an unknown distribution. And I agree with the statement that conventional DRO framework would be problematic if the target distribution is far away from the OOD data. To address this problem, the author further introduces a patch-restricted method to make DRO applicable for OOD settings and effectively improves the performance of learned student models on OOD data. The experiments in table 1 are very interesting. It seems that MosaicKD not only works on similar data like CIFAR10, but also works on extremely mismatched data like SVHN. Besides, extensive experiments are conducted in this work for evaluation. Table 1 benchmarks existing methods on several datasets, which would be helpful for future works.

**[Weaknesses]:**

Here are my questions:
1) To my knowledge, some KD methods are also designed for large-scale unlabeled data, like PU Compression [50]. Why they are not used as baselines in this work?
2) In some settings, data-free KD is also a strong competitor (table 1, resnet34 to resnet18). What is the advantage of the data-driven approach proposed in this paper compared to the data-free approach that does not require data?
3) There is some room for improvement in the selection of patch size, such as adaptive patch partition or selective patch generation.

**[Additional Feedback]:**

In addition to the above issues, there are some suggestions for improvement:
1) Some equations should be presented in more appropriate ways, such as patch distribution in Eqn 3 and the entropy term in Eqn 4.
2) A brief description of the algorithm can be included at the beginning of the Sec. 4.2 to make the whole algorithm clearer.
3) In L205, the citation to the equivalence between GAN objective and JS divergence is missing.

**Update: post rebuttal**

I have read the authors' response and other reviewers' comments. Overall I think the rebuttal addresses most of my concerns and I will increase my rating to *7*.

**Time Spent Reviewing:**

2

---

> ### Author Response · Authors · 2021-08-10
> **Response to Reviewer jgX3**
>
> We sincerely thank the reviewer for the positive comments on our work.
>
> **Q1: Some KD methods are also designed for large-scale unlabeled data, like PU Compression [50]. Why they are not used as baselines in this work**
> **A1:** Thanks for the question. PU compression assumes that there is sufficient  in-domain data for all categories in the unlabeled set and requires original training data for retrieval. As can be seen in Figure 3 of the main paper, many classes are missing in the OOD set, which makes PU infeasible in our settings. Please refer to Section 6 for a discussion about unlabeled KD and the proposed method.
>
> **Q2: What is the advantage of the data-driven approach proposed in this paper compared to the data-free approach that does not require data?**
> **A2:** Data-free methods rely on naive assumptions about data distribution, such as Gaussian assumptions, on intermediate features. They ignore the sophisticated patterns of natural images and usually result in heavily degraded student models. The data-driven method proposed in our work leverages natural patterns for data synthesis, where more complex and plausible samples can be reconstructed. Further, we can merge OOD data and synthetic data to avoid synthesizing the whole transfer set, which is more efficient than data-free KD.
>
> **Q3: Suggestions for improvement: patch size selection, presentation of equations, etc.**
> **A3:** We appreciate the reviewer for the comments. We will improve the strategy of patch size selection and provide empirical studies in our future work. We will also polish the presentation as suggested.

---

### Official Review · Reviewer_LcFs · 2021-07-14

**Rating:** 7
**Confidence:** 4

**Summary:**

This paper proposes a method for knowledge distillation with out-of-domain data (OOD-KD). The key idea is to dismantle out-of-domain images into patches and assemble in-domain data by training a generative adversarial network. It is based on observation that different domain may still share common local patterns from which new (in-domain) categories can be re-assembled. The proposed method is developed undert the framework of the so called Distributionally Robust Optimization (DRO), restricted by a ball space centered at the OOD patch distribution.

**Limitations And Societal Impact:**

The paper conveys the motivation and idea nicely but the presentation needs to be improved. After going through the paper and appendix, I have some questions about the proposed MosaicKD:

1) Adversarial attack adds a small perturbation to the image and is also able to modify the category of OOD data. What is the difference between samples obtained from MosaicKD and adversarial attacks?

2) In Figure 3, why the FID of MosaicKD in some categories is a bit worse than the OOD data?


3) some errors and typos need to be fixed:
- Line 178: The symbol V(G) and V(X’) is a little confusing.
- Line 148: diverged => divergent.
- Line 167: V(X^prime) => V(X^\prime)
- Line 197: The first term refers to … => The second term refers to …
- Line 207,: a redundant bracket for x ~ P_G in Eqn 6.
- Line 221: Equation 4.1 => Equation 4. Besides, I think the derivation of equation 6 and equation 7 should be moved from appendix to the main paper for better presentation.
- Line 298: We conduct ablation … => we conduct ablation study…
- Line 1 in Algorithm 1: D(z; \theta_d) => D(x; \theta_d).


**Main Review:**

Overall, I like the idea of this paper. First, I think the proposed method is a clear and novel way for knowledge distillation from out-of-domain data. Theoretically, the method is reasonable, since it is an extension of DRO framework for distant distributions (OOD and ID data). The empirical evaluation sufficiently validates the authors' claim. I think the proposed method can be useful in practice.

On the other hand, the main weakness of this paper lies in the patch learning on local patterns. Specifically, the proposed method trains a generator on all patches directly and it is not very clear what kind of local patterns is useful for the target domain. Maybe the aligning loss L_{align} can help select task-related patterns for synthesis but is not clear enough in the method part. I think the author should take a further step on this point and provide more discussion on this topic as it is the core of the method.

**Time Spent Reviewing:**

6

---

> ### Author Response · Authors · 2021-08-10
> **Response to Reviewer LcFs**
>
> We sincerely thank the reviewer for the positive comments on our work.
>
> **Q1: What kind of local patterns are useful for the target domain**
> **A1:** Thanks for the question. MosaicKD uses an aligning loss $L_{align}$ as Eqn (4) to select local patterns and assemble them to present desired semantics in the target domain. The aligning loss prefers those task-related patches in the OOD set, which is more helpful for producing high-confidence samples.
>
> **Q2: What is the difference between samples obtained from MosaicKD and adversarial attacks**
> **A2:** Thanks. Adversarial attack fools a teacher model by adding small perturbations to clean images. Typically, these perturbations are unrecognizable for humans but will lead to wrong model predictions. In our method, the entropy term in Equation (4) plays a similar role as adversarial attacks and modifies the teacher's predictions in the same way. We provide empirical study in Figure 1 (a) of the supplemental material, where the entropy term alone fails to exhibit desired semantics.
>
> **Q3: Why the FID of MosaicKD in some categories is a bit worse than the OOD data**
> **A3:** Inevitably, there are some in-domain samples in some categories. These in-domain samples are natural images and thus show better FID scores in some categories than our synthetic one. In addition, such a gap can be also caused by the limited capacity of the generator, as we only use the vanilla GAN architecture in our experiments.
>
> **Q4: Errors and typos**
> **A4:** Thanks for the comments. We will fix these errors in the revised version.

---

### Official Review · Reviewer_yT4j · 2021-07-17

**Rating:** 6
**Confidence:** 3

**Summary:**

This paper considers a new task, knowledge distillation using out-of-domain data, and proposed a novel method, MosaicKD. Leveraging the shared common local patterns, MoscaiKD is able to effectively learn from the OOD data.  The paper is well organized and presented, and experimental results demonstrated the effectiveness of the proposed solution. Even I have some questions about the experiments, the quality of the paper is high.

**Limitations And Societal Impact:**

Yes. No critical points are missing.

**Main Review:**

Originality: In this submission, a novel problem is presented: Out-of-Domain Knowledge Distillation, which could be considered as a simple combination of two sub-problems: domain adaptation and knowledge distillation. To deal with this new problem, a new method MosaicKD is proposed, which also could be considered as a combination of KD and GAN methods (adapted to patches, local patterns). The main idea is presented cleverly, but the contribution is not significant. Combining two sub-problems (DA and KD) and proposing a plausible insight (local pattern)  cloud be a good choice for publication, but not an outstanding work. Related work is adequately cited.

Quality: the submission is technically sound, and the claims are well supported. This work is a complete piece.
Questions about Equations (3) and (4): How can we evaluate the effectiveness of the first term (about the patch) in Equations (3) and (4)? Since the key insight behind this submission is the shared common local patterns, we should validate the effectiveness of the first term. Maybe this could be answered by results in Table 4.?
I understand the insights behind Fig.1, but I could not get helpful information from Fig. 1 as there seems no overlapping local patterns between large images.

Clarity: The submission is clearly written and well organized. The writing is good. Some errors:
Line 197: The first term refers to the entropy …    => should be “The second”?

Significance and  Experiments: Experimental results demonstrate the effectiveness of the proposed methods. However, the comparison is not fair. In Table 1, the authors mainly compared with KD [18] Balanced [35] FitNet [41] RKD [38] methods, which are not relatively new.
In Table 2, first, if the FLOPs indicate the FLOPs of the student model? If so, it is unnecessary to present the FLOPs here since all student models are the same. Also, why use mIoU on ImageNet?
An important experiment is about the patch size (Table 4). We see these datasets using a resolution of 32\times 32 (cifar10 and SVHN, not sure about the Places365). But the best result is achieved given 18 * 18 patch (aka., 1/4 image area) which seems not reasonable.

**Time Spent Reviewing:**

4

---

> ### Author Response · Authors · 2021-08-10
> **Response to Reviewer yT4j**
>
> We sincerely appreciate the reviewer for the constructive comments and suggestions.
>
> **Q1: How can we evaluate the effectiveness of the first term in Equations (3) and (4)?**
> **A1:** We thank the reviewer for this question. We have, in fact, indeed done some ablation studies to verify the effectiveness of the first term in Eqn. (3) and (4). Experimental results are provided in Table 5 in the paper. When the first term is removed (mosaicKD without discrimination, abbr. w/o disc.), the performance of student models degrades severely, dropping by more than 10 percent as compared with the proposed mosaicKD. In addition, we further provide some visualization results in Fig. 1 (a) in the supplementary material, where the first term is removed from Eqn. (4). The synthetic images are visually less authentic and significantly isolated from images in the target domain. If the patch-level loss in Eqn. (4) is replaced with an image-level one, the synthetic samples are trapped in the OOD domain and can hardly be aligned to the target domain. All these results demonstrate the effectiveness of the first term in Eqn. (3) and (4). We will clarify these points in the revised version.
>
> **Q2: I understand the insights behind Figure 1, but I could not get helpful information from Fig. 1, as there seems to be no overlapping local patterns between large images.**
> **A2:**  Thanks for the comment. Despite the domain difference between large images, the overlap between local patterns actually depends on the patch size. As shown in Table 4 in the paper, smaller patches lead to smaller FID between OOD and target data, which indicates that smaller patches cropped from large images share more similar local patterns. However, it may be difficult for us to perceive the improvement on FID based on only a small number of samples in Figure 1.
>
> **Q3: Some errors: Line 197: The first term refers to the entropy … => should be “The second”?**
> **A3:** Thanks for pointing out this issue. We will fix this error in the revised version.
>
> **Q4: The authors mainly compared with KD, Balanced, FitNet, RKD methods, which are not relatively new. The comparison is not fair in Table 1.**
> **A4:** Our work focuses on answering the question of "what type of data is useful for KD", which is orthogonal to the current literature where the focus is usually on "how to transfer knowledge given a pre-defined dataset". Thus, we used vanilla KD to prove the effectiveness of the data generation in MosaicKD.
>
> As suggested, we have also compared our method to the latest KD methods like CRD [1] and SSKD [2] on CIFAR-100, using CIFAR-10 as OOD data. The results are listed as follows.
>
> | Method  |  resnet34 => resnet18  | vgg11 => resnet18 | wrn40-2 => wrn16-1 | wrn40-2 => wrn40-1 | wrn40-2 => wrn16-2 |
> |:------:|:------:|:------:|:------:|:------:|:------:|
> |  KD      | 73.55 | 68.04 | 47.47 | 61.17 | 63.48 |
> | CRD    | 71.23 | 66.48 | 47.00 | 59.59 | 61.37  |
> | SSKD  | 73.81 | 68.72 | 49.57 | 60.71 | 64.61  |
> | Ours    | 77.01 | 71.56 | 61.01 | 69.14 | 69.41 |
>
> It can be seen that the results of these SOTA distillation methods, still, barely match those of the proposed method, which implies the OOD nature of the training data is the main obstacle for distillation. We will add the SSKD results to the main paper in the revised version.
>
> [1] Contrastive Representation Distillation, Yonglong Tian et al., ICLR'2020
> [2] Knowledge Distillation Meets Self-Supervision, Guodong Xu et al., ECCV'2020
>
> **Q5: It is unnecessary to present the FLOPs here since all student models are the same.**
> **A5:** Thanks for the comment. We will replace the FLOPs of students with the amount of trainable parameters for data synthesis. The results are listed as the following:
>
> |Teacher|Student|ZSKT|DAFL|KD|Ours|
> |:------:|:------:|:------:|:------:|:------:|:------:|
> | - (No synthesis) | - | 2.02M (G) | 2.02M (G) | - | 2.02M (G)+0.53M (D) |
>
> where (G) and (D) refers to the generator and discriminator. Our method does not introduce too much training effort as compared to existing data-free KD methods.
>
> **Q6: Why use mIoU on ImageNet**
> **A6:** We are sorry for the misunderstanding. In fact, ImageNet is only used as the OOD training data. The evaluation is conducted on NYUv2, a popular benchmark for segmentation.
>
> **Q7: The best result is achieved given 18x18 patch which seems not reasonable**
> **A7:** Thanks. CIFAR10 results in table 4 showcase that both 8x8 and 18x18 are viable for MosaicKD, and their difference is truly negligible (0.02\%). The patch size is a trade-off hyper-parameter between the domain gap (FID) and the structural pattern. As mentioned in L168, excessively small patches will lead to less structural patterns and intractable searching space for DRO, while over-sized patches result in large domain gaps. As can be seen from Table 4, our method is robust to a wide range of patch sizes if OOD and ID data are similar (CIFAR-10 & CIFAR-100), but may be sensitive for dissimilar ones (SVHN & CIFAR-100).

---

### Decision · Program_Chairs · 2021-09-27

**Decision:**

Accept (Poster)

**Comment:**

The paper studies a new setting, called out-of-domain knowledge distillation, where the teacher network and the student network are trained on different datasets. The proposed new setting is interesting and more practical than existing ones. In terms of the solution, the authors propose a mosaic idea to synthesize in-domain data by imitating the local patterns from real world OOD data.  The key technical idea is to use minimax optimization to ensure the synthesized data could fool the discriminator. The solution is novel and technically sound. Reviewers all agreed that the proposed problem is interesting and the solution is novel.Thus, I am recommending acceptance.

Of course, the authors need to carefully revise the manuscript according to reviewers’ comments. The answers to Q3 and Q4 of Reviewer yT4j must be included in the final version.